# In Situ Photopolymerization of Acrylamide Hydrogel to Coat Cellulose Acetate Nanofibers for Drug Delivery System

**DOI:** 10.3390/polym13111863

**Published:** 2021-06-03

**Authors:** Mohamed F. Attia, Ahmed S. Montaser, Md Arifuzzaman, Megan Pitz, Khouloud Jlassi, Angela Alexander-Bryant, Stephen S. Kelly, Frank Alexis, Daniel C. Whitehead

**Affiliations:** 1Department of Chemistry, Clemson University, Clemson, SC 29634, USA; marifuz@g.clemson.edu; 2Division of Pharmacoengineering and Molecular Pharmaceutics, Center for Nanotechnology in Drug Delivery, Eshelman School of Pharmacy, University of North Carolina at Chapel Hill, Chapel Hill, NC 27599, USA; 3Textile Research Division, Pretreatment and Finishing Department, National Research Center, Dokki, Cairo 12622, Egypt; 4Department of Bioengineering, Clemson University, Clemson, SC 29634, USA; mpitz@g.clemson.edu (M.P.); angelaa@clemson.edu (A.A.-B.); 5Center for Advanced Materials, Qatar University, Doha P.O. Box 2713, Qatar; khouloud.jlassi@qu.edu.qa; 6Department of Forest Biomaterials, College of Natural Resources, North Carolina State University, Raleigh, NC 27607, USA; sskelley@ncsu.edu; 7School of Biological Sciences and Engineering, Yachay Tech University, Urcuqui 100650, Ecuador; falexis@yachaytech.edu.ec

**Keywords:** electrospinning, cellulose acetate, nanofibers, poly(acrylamide) hydrogel, drug delivery, kinetic release

## Abstract

In this study we developed electrospun cellulose acetate nanofibers (CANFs) that were loaded with a model non-steroidal anti-inflammatory drug (NSAID) (ibuprofen, Ib) and coated with poly(acrylamide) (poly-AAm) hydrogel polymer using two consecutive steps: an electrospinning process followed by photopolymerization of AAm. Coated and non-coated CANF formulations were characterized by several microscopic and spectroscopic techniques to evaluate their physicochemical properties. An analysis of the kinetic release profile of Ib showed noticeable differences due to the presence or absence of the poly-AAm hydrogel polymer. Poly-AAm coating facilitated a constant release rate of drug as opposed to a more conventional burst release. The non-coated CANFs showed low cumulative drug release concentrations (ca. 35 and 83% at 5 and 10% loading, respectively). Conversely, poly-AAm coated CANFs were found to promote the release of drug (ca. 84 and 99.8% at 5 and 10% loading, respectively). Finally, the CANFs were found to be superbly cytocompatible.

## 1. Introduction

Over the past few decades, nanofibers have shown immense potential in a broad range of applications such as energy storage [1,2,3], environmental remediation [4], agriculture [5], and filtration [6]. More relevant to this study, they are also useful in a variety of biomedical applications including biosensing [7], diagnosis [8,9], drug delivery [10,11,12], wound dressing [13,14,15], regenerative medicine [16,17], and scaffold tissue engineering [18,19,20]. Nanofibers have been mostly produced by electrospinning [21,22], in which polymeric solutions are formed into nano-sized continuous fibers by applying a strong electric field. Therein, the polymer solution is squeezed from a syringe into a drop. High electrostatic forces overcome the cohesive forces, resulting in formation of the jet, its continuous elongation under a whipping motion, evaporation of the solvent, and formation of the nanofibers [23,24].

Electrospinning can be classified based on electrospinning process, solution, and environmental parameters. (i) The electrospinning parameters encompass the applied electric field, needle type (i.e., single, double or multiple needles, coaxial, tri-axial, multi-axial needles), needle diameter, flow rate, and distance between the needle and collector. (ii) The solution parameters include the identity of the solvent, polymer concentration, viscosity, and solution conductivity. (iii) The environmental parameters include relativity humidity and temperature. All of these parameters directly affect the generation of smooth and bead-free electrospun fibers.

As compared with other polymer nanofiber manufacturing technologies (i.e., phase separation method or self-assembly method), electrospinning technology benefits from simple operation parameters and low cost [25]. Electrospinning is a straightforward and versatile way to generate electrospun fibers with different morphologies with diameters ranging from the nano- to microscale [26,27,28]. Some strategies have also been presented to generate 2D [29] and 3D fiber assemblies by arranging 1D nanofibers [30].

Interestingly, electrospinning is capable of producing nano and microfibers with relatively high surface area-to-volume ratios as compared to other membranes developed by conventional techniques. The technique offers the possibility of synthesizing fibers from mixed polymers with controlled morphologies [31], as well as engineering core-shell and coaxial electrospun fibers from two different polymer solutions [32]. Multilayers of different polymers are accessible through a “layer-by-layer” strategy [33] by means of electrostatic, physical, or chemical interactions.

Electrospun nanofibers have been explored in biomedical science for applications encompassing drug delivery systems, diagnostic imaging, theranostics, and tissue engineering [34]. They are well suited for these applications because they are readily engineered for specific applications by controlling their structures and properties such as porosity, diameter, stacking, alignment, patterning, surface functional groups, biodegradability, and mechanical properties. Both 2D and 3D scaffolds have been generated to control cell migration and stem cell differentiation for improving the regeneration/repair of various types of tissues (i.e., skin, nerve, heart, and musculoskeletal system) and tissue interfaces [35]. Of particular interest, they have been found to be efficient targeted therapeutic carriers for delivering nucleic acids [36,37], proteins [38], microorganisms [39], and stem cells [40], as well as means to control the release profile of bioactive molecules [41,42]. Electrospinning allows for the facile loading of drugs within the nanofibers platform through mixing drugs with a polymer solution, resulting in electrospun fibers with relatively high drug loading capacity. The choice of the polymer employed in the spinning process is usually dictated by the solubility profile of the drug [43]. For example, water-soluble drugs have commonly been spun with polyvinyl alcohol polymers [44], while organic-soluble drugs are usually spun with cellulose acetate or polystyrene polymers [45]. Typically, the electrospinning performance of organic soluble polymers is better than water soluble solutions, because the lower boiling points of most organic solvents allows for rapid solvent evaporation as compared to water.

One major concern confronting drug delivery systems prepared by electrospinning techniques is the potential for fast dissociation of the drug from the surface layer of the nanofibers, a phenomenon known as burst release [46]. To overcome this challenge, drugs can be coated with monodispersed core/shell particles with the outer layer of polymer. Core/shell composite nanofibers of drugs and bioactive ingredients can be synthesized using coaxial electrospinning [47]. However, in some cases, coaxial electrospinning does not apply for all polymers to produce desirable nanofibers. Consequently, we used a two-step based methodology in this study.

Cellulose acetate (CA), an organic-soluble polymer derived from cellulose through an acetylation process, shows potential for electrospinning based on several favorable parameters including, but not limited to, facile spinning process, stability, scalability, and reinforcement of membrane physical properties. Cellulose acetate is occasionally mixed with other polymers to improve mechanical properties of the resulting materials [48].

In the present work, we developed a simple strategy using electrospinning to coat cellulose acetate nanofibers (CANFs) loaded with a poorly water-soluble model drug (ibuprofen, 5 and 10 wt.%) with poly-AAm hydrogel to confine/embed the drug in the nanofiber textures and thus preventing burst release. Additionally, we thought that this approach might control drug release from the nanofiber and significantly facilitate the diffusion of higher amounts of drug bonded chemically or physically to the nanofibers. The choice of poly-AAm hydrogel as a coating agent was motivated by its safety, biocompatibility, biodegradability, high water absorbability, potential for cross-linking, and its efficiency as a drug nanocarrier. The engineered nanofibers also showed excellent cytocompatibility in assays using 3T3 adipose cells in vitro. The resulting data suggest a promising performance of the material for potential applications in wound dressing.

## 2. Experimental Section

### 2.1. Materials

A neutral monomer, acrylamide (AAm) (≥99%, d = 1.13 g mL^−1^), cross-linker (methylene bisacrylamide (MBAAm)), the UV initiator (2,2-dimethoxy-2-phenylacetophenone (DMPA), 99%), cellulose acetate (CA, Mw = 30 kDa, d = 1.3 g mL^−1^), non-steroidal anti-inflammatory drug (ibuprofen (Ib, ≥98%), sodium chloride (NaCl), potassium chloride (KCl), disodium hydrogen phosphate (Na_2_HPO_4_), potassium dihydrogen phosphate (KH_2_PO_4_), *N*,*N*′-dimethyl formamide (DMF), dimethylformamide-dimethyl acetal (DMF-DMA), and acetone were purchased from Sigma-Aldrich (St. Louis, MO, USA) and used as received. Distilled water was used in all experiments. To construct a polymer coating synthesis reaction cell, a pair of flint glass plates, silicone rubber film, and binder clips were used. For cell experiments, 3T3 Adipose cell lines were purchased from the American Type Culture Collection (ATCC, Manassas, VA, USA). Fetal bovine serum (FBS), phosphate buffered saline (PBS), Dulbecco’s modified Eagle medium (DMEM), and the antibiotics penicillin/streptomycin were obtained from PAN Biotech (Aidenbach, Germany). Dimethyl sulfoxide (DMSO) and 3-(4,5-dimethylthiazol-2-yl)-5-(3-carboxymethoxyphenyl)-2-(4-sulfophenyl)-2*H*-tetrazolium) (MTS) solutions were purchased from Sigma-Aldrich (St. Louis, MO, USA).

### 2.2. Preparation of Ibuprofen-Loaded CANFs and Coating with Hydrogel Polymer in Two Consecutive Steps

#### 2.2.1. Synthesis of Electrospun CANFs Loaded with Ibuprofen

An electrospinning technique was utilized to fabricate CANFs. Cellulose acetate (18 wt/v%) was solubilized in dimethyl acetamide/acetone (2:1) solution and loaded with different concentrations of drug (5 wt.% and 10 wt.% of Ib proportional to cellulose acetate content). These two formulations were separately taken up in a 5 mL glass syringe and connected to a metal capillary needle with an exit orifice diameter of 0.6 mm. The electrospinning apparatus was equipped with a high voltage of 16 kV (Tianjing High Voltage Power Supply Co., Tianjing, China), and the flow rate of the polymer solution out of the syringe and through the capillary was controlled at 2 mL h^−1^ by a microinjection pump. The produced CANFs loaded with drug were collected on a rotating collector covered with aluminum foil at a distance of 15 cm before drying under vacuum. For comparison, non-loaded CANFs were also prepared as a control sample following the same procedure without drug.

#### 2.2.2. In Situ UV Polymerization of AAm Hydrogel Coating on Drug-Loaded CANFs

To coat the drug-loaded nanofiber film, a polymerizable solution was prepared by mixing the neutral monomer, acrylamide (AAm) (15 wt.%), the chemical cross-linker, methylene bisacrylamide (MBAAm) (5 wt.%), and the UV initiator, 2,2-dimethoxy-2-phenylacetophenone (DMPA) (0.75 wt.%) together in *N*,*N*′-dimethyl formamide (DMF) solvent. Finally, a clear solution was obtained and degassed by bubbling with N_2_ for a few minutes and kept in a glass vial under an inert environment to use as a coating solution. Next, a 100 mg sample of nanofiber film was taken on a pool prepared on a cleaned glass substrate and fully soaked with the polymerizable solution. The mass of the polymerizable solution was ca. 7× higher than the mass of the nanofiber film. The concentration of drug in the nanofiber films was varied at 5 and 10 wt.%. Afterward, under an inert nitrogen atmosphere maintained in a Cleatech^®^ 2100-4-C glove box (Cleatech, LLC, Santa Ana, CA, USA) with attached oxygen analyzer and a Cleatech^®^ A21-HM-OA Nitrogen Purge controller, a photo-polymerization reaction was performed by applying UV radiation (365 nm) from the top to the nanofiber film fully immersed in the reaction solution at approximately 45 °C for 2 h. After completion of the polymerization, a uniform rectangular (ca. 20 mm × 10 mm rectangle, and ca. 0.5 mm thick) film coated with poly-AAm was obtained. The nanofiber films coated with poly-AAm were removed from the reaction substrate and air-dried. Later, the dried and coated nanofibers were characterized and compared with uncoated nanofibers and tested by measuring drug release rate.

### 2.3. Characterization of Nanofibers

Fourier transform infrared (FT-IR) spectra were collected from dried samples of nanofibers which were placed directly in a Nicolet NEXUS870 spectrometer (Nicolet Instruments Inc., Madison, WI, USA). All spectra were collected with a 2 cm^−1^ resolution after 32 continuous scans. Thermogravimetric analysis (TGA) was conducted on dried samples using a TA Instruments Hi-Res TGA 2950 thermogravimetric analyzer (Waters/TA Instruments, New Castle, DE, USA) over the temperature range of 25 to 600 °C at a rate of 20 °C min^−1^ under nitrogen flow. Powder X-ray diffraction (XRD) was used to determine whether the materials constituted nanofibers using a Rigaku Ultima IV diffractometer with Cu Kα radiation (Rigaku, Tokyo, Japan). All materials (*i.e.,* CANFs, CANFs/Ib, CANFs/Ib/PAA) were analyzed on a zero-background sample holder. Data were collected between 5° and 65° 2θ at a scan rate of 1 degree per minute in 0.02-degree intervals. Scanning Electron Microscopy (SEM) was conducted using a S6600 variable pressure SEM (Hitachi, Tokyo, Japan) equipped with an in-lens detector to check the morphology of uncoated and coated CANF/Ib with poly-AAm. Ultraviolet-visible (UV-Vis) absorbance spectra were recorded on a UNICAM HELIOS UV-Vis spectrophotometer (Varioskan Flash, Thermo Scientific, Waltam, MA, USA). Absorbance measurements were carried out on ibuprofen drug in supernatants with maximum wavelength of 264 nm to quantify the drug released from the nanofibers.

### 2.4. In Vitro Drug Release Kinetics

The dialysis bag diffusion method was used to analyze the in vitro drug release of ibuprofen from CANFs, which was investigated in phosphate-buffered saline at pH 7.4. Briefly, 50 mg CANF samples were immersed in 10 mL PBS in a regenerated cellulose dialysis bag (MWCO, 12000–14000, Sigma, St. Louis, MO, USA). The closed bag was then immersed into a beaker containing 100 mL of release medium, PBS at pH = 4. The system was then stirred at 120 rpm at 37 ± 0.5 °C. At predetermined time intervals, 1 mL of release medium was removed for UV-Vis spectroscopic analysis at 264 nm to determine the concentration of the released drug. The aliquot was then replaced with 1 mL of fresh PBS solution. Each batch of experiments was performed in triplicate. The cumulative releases of drug from CANFs were plotted against time and the results were reported as mean ± standard deviation. The second-order derivative of the UV spectra was obtained to determine the amount of drug released. A calibration curve of Ib at pH of 7.4 was used.

### 2.5. In Vitro Cellular Studies

#### 2.5.1. Ethylene Oxide (EtO) Sterilization of Nanofiber Samples for Cellular Experiments

Samples with several weights were placed in a 24-well tissue culture plate and sterilized by an Anprolene sterilizer using EtO gas (Model AN74.64, Andersen Sterilizers, Inc., Haw River, NC, USA) with a cycle of 12 h. All samples were placed in breathable packaging that allows for gas to penetrate the sterile barrier and reach the surface of the samples.

#### 2.5.2. Cell Cultivation Protocol

Cells were cultured in Dulbecco’s Modified Eagle Medium (DMEM) (Gibco, Waltam, MA, USA) supplemented with 10% fetal bovine serum (HyClone, Marlborough, MA, USA), 2 μM l-glutamine, 100 μg/mL streptomycin, and 100 U/mL penicillin (Gibco, Waltam, MA, USA) at 37 °C in a 5% CO_2_ humidified atmosphere. Once the cells reached ca. 80–90% confluence, they were trypsinized; the subcultivation ratio was 1:6 to 1:8.

#### 2.5.3. Cytotoxicity Measurements (MTS Assay)

First experiment—direct MTS assay: 3T3 cells were seeded into 96-well plates at a density of 10,000 cells per well. Cells were allowed to attach during an overnight incubation. After cell attachment, nanofibers were added to each well at either 2 mg/mL, 4 mg/mL, or 8 mg/mL concentrations in triplicate. Cells and nanofibers were incubated together for 24 h. Following the 24 h incubation, an MTS assay was used to determine cell viability.

Second experiment—reverse MTS assay: 3T3 Adipose cells were seeded in a 24-well plate onto different weights of nanofiber at a density of 4 × 10^3^ cells per well and were incubated overnight. On days 1, 3, and 5, 100 μL of media was removed from each well and incubated with MTS solution in another 96-well plate for 4 h. The wells were subsequently washed with PBS and replaced with fresh DMEM media. The MTS assay was completed by reading absorbance at 490 nm on a BioTek plate reader (BioTek U.S., Winooski, VT, USA). The rationale here is that the dead cells floated in the media, but the live cells were adhered on the surface of the nanofibers. Thus, dead cells were quantified and then subtracted from the whole cell population to calculate the number of live cells.

## 3. Results and Discussion

Synthesis of electrospun nanofibers: While the use of cellulose acetate nanofibers as a potential therapeutic carrier has been explored previously, as highlighted above, challenges still remain. Specifically, burst release or uncontrolled release of the therapeutic from the nanofibers are challenges that still must be overcome. To explore strategies to mitigate these problems, we loaded an anti-inflammatory drug, ibuprofen, onto the CANFs by electrospinning. Next, we coated a poly-AAm polymer layer on the surface of the loaded CANFs to address the shortcomings associated with poor drug release behavior that is observed with conventional methods for nanofiber-based drug delivery applications such as large-burst drug release, uncontrolled duration of drug release, and incomplete drug release [46,49]. We first attempted to generate the hydrogel polymer coating layer on the CANFs in one step via coaxial spinning, however due to its water solubility, the hydrogel could not be directly applied to electrospinning and led to the production of undesirable nanofibers when mixed with another organic soluble material. Therefore, we decided to prepare the coated CANFs via a two-step process, fabricating the CANFs followed by a UV polymerization coating process. We initially solubilized cellulose acetate in acetone, however, due to the low boiling point of the solvent, this approach caused spinneret closure during the spinning process. Consequently, a mixture of dimethyl acetamide/acetone (DMAc/Ac; 2:1) was used to increase the boiling point of the organic solvent in order to facilitate the electrospinning process. We incorporated 5 and 10 wt.% ibuprofen (i.e., with respect to cellulose acetate) into the organic phase to fabricate drug-loaded nanofibers. Thereafter, the nanofibers were dried under vacuum and then immersed in a pool of polymerizable solution containing the neutral monomer, AAm (15 wt.%), chemical cross-linker, MBAAm (5 wt.%), and the UV initiator, DMPA, (0.75 wt.%). This mixture was then subjected to UV irradiation at 365 nm for 2 h at 45 °C (Figure 1). The developed drug-loaded nanofibers coated with poly-AAm hydrogel were dried under vacuum prior to analysis.

Fourier Transform Infrared (FT-IR) spectroscopy: Figure 2a presents FT-IR spectra of the parent CANF, ibuprofen, CANF loaded with 5 and 10 wt.% ibuprofen, and CANF coated with poly-AAm polymer. The parent CANF spectrum indicated an absorption peak at 3473 cm^−1^ corresponding to the stretching of intermolecular hydrogen bonds of the hydroxyl groups (−OH). The C-H stretching of methyl groups (–CH_3_) was assigned to 2940 cm^−1^. The carbonyl group (C=O) stretching appeared at 1735 cm^−1^. The characteristic peaks of (C–O–C) anti-symmetric stretching vibrations of the ester group of pure cellulose acetate were observed at 1222 cm^−1^. The (C-OH) stretching vibration was noted at 1031 cm^−1^. Three characteristic peaks appeared in the spectrum for ibuprofen: OH group at 2952 cm^−1^, carbonyl group at 1708 cm^−1^, and COO group at 1506–1378 cm^−1^. In the spectrum of drug-loaded CANF (5 and 10 wt.% loading), peaks corresponding to both materials were detected, similar to those indicated for the parent CANF and drug independently, demonstrating the integration of the drug within nanofiber architectures. After coating with poly-AAm hydrogel polymer, new absorption peaks were noted: two signals at 3330 and 3190 cm^−1^, which were assigned to the NH_2_ group, an absorption peak at 1648 cm^−1^ corresponding to CO–NH_2_, and an absorption peak with low intensity at 1745 cm^−1^, which resulted from both the drug and nanofiber. These data confirm the successful coating process of the nanofiber/drug with hydrogel polymer.

Thermogravimetric (TGA) analysis: TGA curves (Figure 2b) also confirm the composition of the materials integrated into the nanofibers. First, the thermal degradation of pure ibuprofen was shown to be a one-stage decomposition beginning at 120 °C with a maximum weight loss (100 wt.%) at 223 °C. Secondly, the thermal degradation profile of pure CANF exhibited a two-stage decomposition: a 3% weight loss occurred between 25 and 137 °C due to evaporation of adsorbed water within the nanofiber. A second stage decomposition occurred between 153 and 465 °C, which accounted for 84.5% weight loss. Third, with 5 and 10 wt.% drug loaded onto the nanofibers, both curves exhibited a similar degradation profile comprising a three-stage decomposition wherein the initial 2.5% weight loss occurred between 25 and 147 °C due to the release of adsorbed water for both drug concentrations. The second stage of decomposition began between 150 and 244 °C, resulting in a weight loss of 6.2 and 9.7% for 5 and 10 wt.% drug loading, respectively. This second stage of decomposition is likely due to the release of drug from the nanofiber upon heating. The third thermal degradation stage occurred between 244 and 434 °C producing a weight loss of 74.6 and 78% for 5 and 10 wt.% drug loading, respectively. This final stage corresponds to the decomposition of the CANF itself. The difference in the weight loss profiles for the two formulations arises from the two different drug concentrations that were deposited onto the nanofiber during the electrospinning process. Fourth, drug-loaded CANF coated with hydrogel polymer exhibited three thermal decomposition stages; (i) a weight loss of 6% detected between 25 and 109.5 °C attributed to the desorption of water, (ii) a second weight loss of 4.2% between 124 and 227 °C, presumably due to degradation of drug and coated hydrogel polymer, and (iii) a third significant weight loss of 70.4% between 261 and 504 °C due to the decomposition of the CANFs. In summary, the FT-IR spectroscopy and TGA analysis demonstrated the successful coating of nanofibers/drugs with hydrogel polymer.

Powder X-ray diffraction (PXRD): We also evaluated the powder patterns of the materials to probe the coating process. We compared the XRD data for all developed nanofibers with two controls (i.e., ibuprofen and CANFs) (Figure 3). The XRD pattern of pristine electrospun cellulose acetate nanofibers displayed two typical broad peaks at 2θ = 9.5 and 19˚ elucidating its amorphous structure. For ibuprofen, the XRD pattern shows high crystalline peaks, which matched previously reported patterns in the literature [50]. On the other hand, when loading the drug with different concentrations on the nanofiber, the resulting XRD pattern is similar to that of pristine cellulose acetate material and the drug loses its crystallinity. As anticipated, no diffraction peaks for ibuprofen were recorded for the drug-loaded CANFs, suggesting that ibuprofen molecules are dispersed inside the CANFs and the inclusion complexation prevents the crystallization of drug molecules. This could also be likely due to a chemical interaction between drug and CANFs during the solution and electrospinning processes. It is important to note that the amorphous state of drug is preferred because it facilitates fast dissolution. After the drug-loaded CANFs were coated with poly-AAm, only a broad peak at 2θ = 22.7° was noted, while no peaks were detected for the drug-loaded CANFs, suggesting the complete coverage of the CANFs with the hydrogel polymer.

Scanning electron microscopy (SEM): The morphology of the nanofiber formulations was evaluated by SEM (Figure 4). The uncoated, electrospun CANFs with and without the drug appeared to be nearly identical, exhibiting a diameter of ca. 25 nm with no beads formed within the nanofiber matrix. This suggested that loading 5 and 10 wt.% of the drug on the CANFs had no detectable impact on the morphology of the material. Conversely, the polymerization process and coating of the CANFs with poly-AAm hydrogel polymer significantly affects crystallinity of the resulting product by creating an amorphous layer on the surface of the CANFs. This change in morphology confirms the successful coating of the CANFs through in situ polymerization of the AAm monomer forming the poly-AAm polymer hydrogel to entrap the drug-loaded CANFs. The SEM analysis is consistent with the XRD data discussed above.

Drug release profile: Since the release of drug molecules occurs primarily by diffusion, it has been demonstrated that the release profile of bioactive molecules from electrospun fibers can be influenced by biodegradability, hydrophobicity, hydrophilicity, fiber diameter, and configuration [51]. As previously reported, electrospun biodegradable PCL/collagen scaffolds were synthesized under conditions in which the solution concentration, needle diameter, flow rate, and distance from the substrate were carefully controlled to provide various fiber diameters exhibiting different drug release properties [52]. It is also known that the hydrophilicity of protein-carrier materials based on electrospun fibrous poly(lactide-co-glycolide) (PLGA) can actively influence the release kinetics of the protein released from the scaffolds [51]. Further, in their study, Okuda et al. concluded that the fiber diameter impacts the release profiles of drug-loaded electrospun fibers, where smaller fibers exhibited rapid drug release in the initial stage [53].

Due to the loss of some drug from the samples into the reaction medium during the coating process, we determined the loaded drug on the nanofibers for both concentrations again. After complete drying, the NFs were dissolved in ethanol for 48 h with stirring to completely release all loaded ibuprofen, which was quantified by UV-Vis spectroscopy. We found for the 5% drug-loaded sample, the drug was reduced by 3.1% to achieve 96.9% loading efficiency, while for the 10% drug-loaded formulation, 4.8% drug was eliminated to achieve 95.2% loading efficiency. In this study we found that the coating agent also impacts the release profile (Figure 5). The advantage of using poly-AAm hydrogel to coat the drug-loaded CANFs was the resulting ability to increase the amount of drug released from the CANFs (i.e., an increase from ca. 35% to 84.2% cumulative release for the 5 wt.% Ib-loaded CANFs and an increase from ca. 83% to 99.8% cumulative release for the 10 wt.% Ib-loaded CANFs). It should be noted that the resulting release profile variation before and after coating at 5% drug NFs is higher than that of 10% drug NFs. This could be related to the drug/hydrogel ratio of the formulation, because we added the same hydrogel concentration with both 5 and 10% drug samples. Furthermore, the hydrogel coating increases the hydrophilicity of the nanofibers, which results in a higher release rate of nanofiber coated with hydrogel compared to nanofibers alone when the loading is 5%. However, when the loading is 10%, the effect of the hydrogel coating is not observed because it is expected that a higher loading of drug into the nanofibers will result in a burst release of the drug. When the drug ratio increases in the nanofiber, the release rate increases due to the burst release [54,55].

The entire release of drug cargo from both poly-AAm coated CANF formulations occurred after 8 h, exhibiting similar constant release rate over time. The conventional release profile of ibuprofen facilitates the rapid absorption with a relatively short elimination half-life of about 2 h. Therefore, the drug is typically administered three or four times daily to maintain the therapeutically effective plasma concentrations over 24 h [56]. The sustained release profile of ibuprofen from poly-AAm coated CANFs offers a significant improvement on this scenario in terms of drug delivery.

In vitro cytotoxicity assay: Since NSAIDs have previously been reported as stimulants for adipogenesis, which might also be appropriate for adipose tissue engineering, we adopted adipose cells as a model for toxicity measurements [57]. An MTS assay with 3T3 adipose cells was conducted to evaluate the potential cytotoxicity of the CANF formulations (Figure 6). The raw data from the MTS assay was normalized with respect to the viability of cells grown in wells without nanofibers, thus the results are presented as percent viability relative to cells grown without nanofibers. At all concentrations, the blank nanofibers do not cause cytotoxicity. At 2 mg/mL, all nanofiber treatments retain at or above 100% cell viability. However, at 4 and 8 mg/mL, drug loaded nanofibers do result in a decrease in cell viability.

Further, we studied the cytotoxicity of all nanofibers over time, as presented in Appendix A) Using 4 mg of CANFs in 1 mL of culture media, cell viability initially decreases with respect to untreated controls. However, viability at day 5 of treatment is above 80% for each CANF formulation, indicating that the materials display acceptable levels of cytocompatibility. For nanofibers dosed at a concentration of 8 mg/mL of culture medium, slightly higher cytotoxicity is observed. The PAAm coated, 10 wt.% Ib-loaded CANFs (NF10 + PAA) showed the lowest cytotoxicity, with percent viability remaining above 80% through day 5 of treatment. These results indicate that employing the CANFs at a concentration of 4 mg/mL may be more beneficial, with the overall viability increasing from day 3 to day 5, while CANF formulations at a concentration of 8 mg/mL may be too cytotoxic since viability decreases throughout the 5-day experiment. Overall, the developed CANFs exhibited excellent biocompatibility, validating their use in vivo.

## 4. Conclusions

In this study, we investigated the effects of poly-AAm hydrogel coating on the drug release profile of cellulose nanofibers (CANFs) loaded with the NSAID, ibuprofen. The preparation of these materials was achieved by a two-step process: formulation of electrospun CANFs loaded with the drug followed by coating with poly-AAm hydrogel polymer by means of an in situ polymerization approach. Both uncoated and coated CANFs were characterized by FT-IR, TGA, PXRD, and SEM analyses to investigate their chemical composition and morphology. Poly-AAm coating of the drug-loaded CANFs was found to change the release profile of loaded ibuprofen by controlling the release rate and preventing undesired burst release. The drug loading of ibuprofen has significant effects on the CANFs but not on the CANFs hydrogel matrix. The CANF formulations were also found to exhibit excellent biocompatibility with 3T3 adipose cells in vitro, demonstrating that the hydrogel did not affect the toxicity. These findings demonstrate the potential of CANF/drug complex coated with hydrogel polymer as safe, effective drug nanocarriers and promising drug delivery matrix.

## Figures and Tables

**Figure 1 polymers-13-01863-f001:**
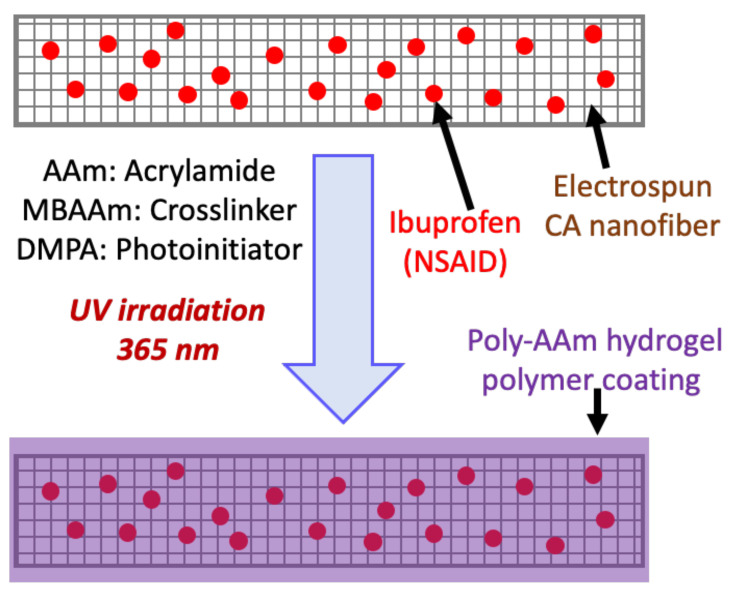
Schematic representation indicating the fabrication of CANFs loaded with ibuprofen drug and coated with poly-AAm hydrogel via two consecutive steps.

**Figure 2 polymers-13-01863-f002:**
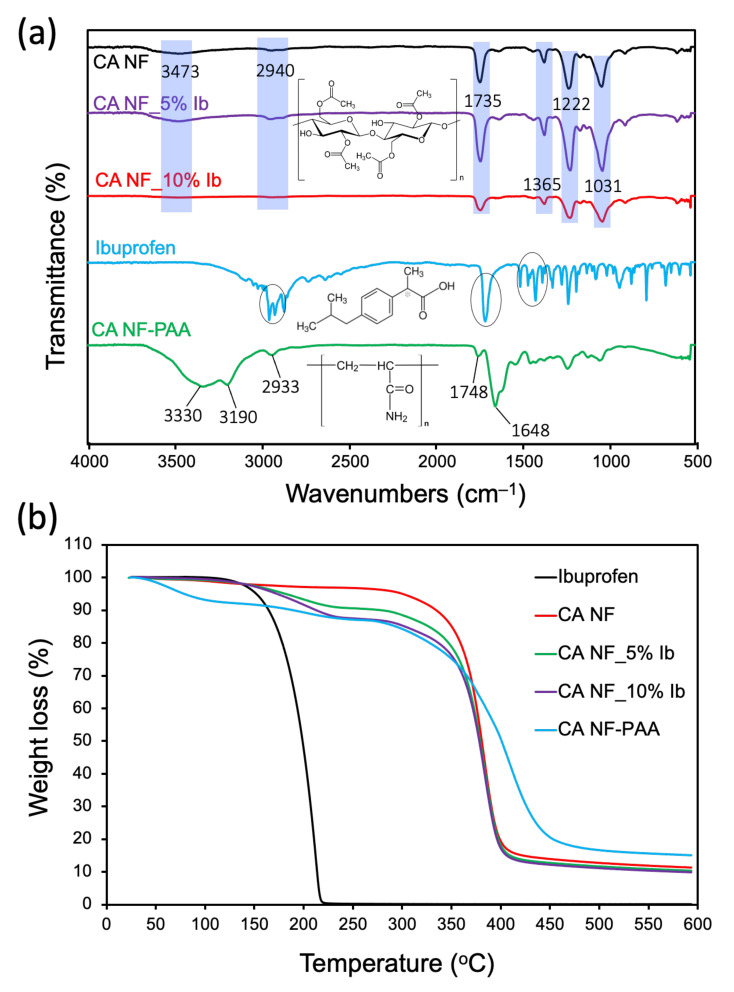
(**a**) FTIR spectra analyzed between 4000 and 500 cm^−1^ and (**b**) TGA analysis for all formulated nanofibers including unmodified CANF, CANF loaded with 5 and 10 wt.% ibuprofen and drug-loaded CANFs coated with poly-AAm hydrogel.

**Figure 3 polymers-13-01863-f003:**
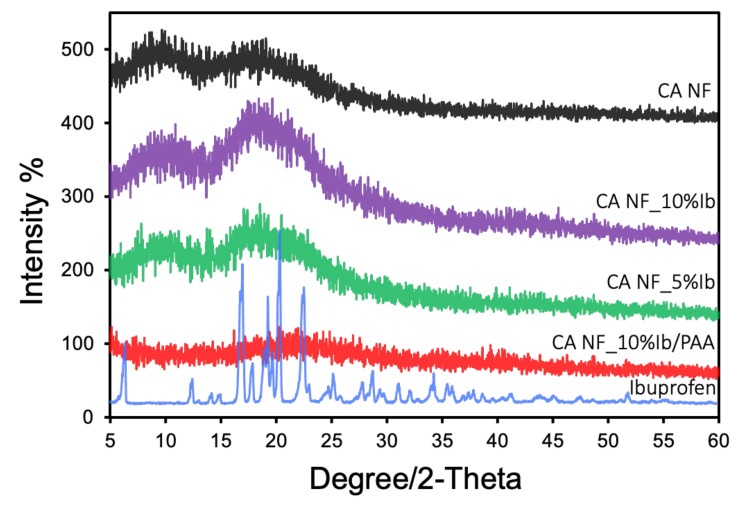
XRD patterns of ibuprofen, pristine CANFs, CANFs loaded with 5 and 10 wt.% ibuprofen, and drug-loaded CANFs coated with poly-AAm.

**Figure 4 polymers-13-01863-f004:**
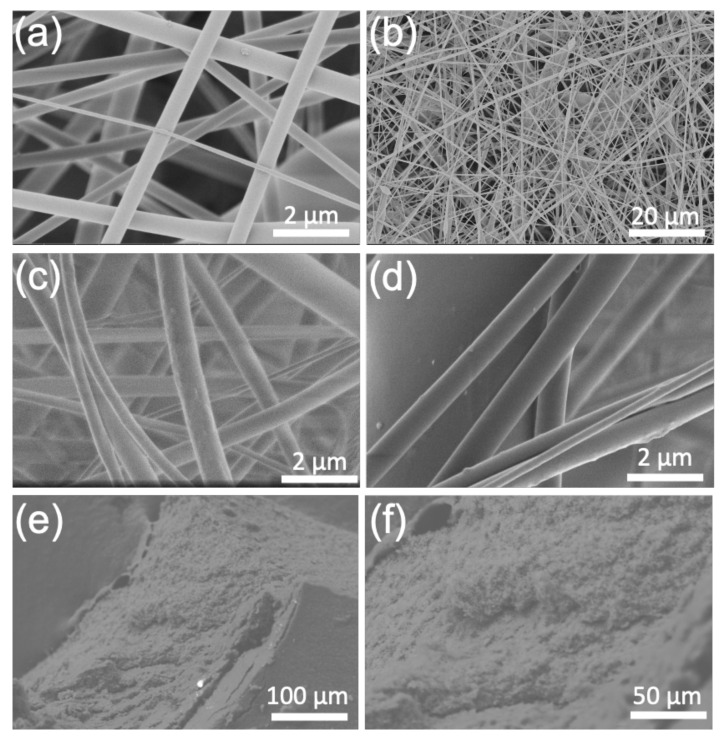
(**a**,**b**) SEM micrographs of unmodified CANFs (blank), (**c**) CANFs loaded with 5 wt. % Ib (**d**) CANFs loaded with 10 wt.% Ib, (**e**) 5 wt.% Ib-loaded CANFs coated with poly-AAm, and (**f**) 10 wt.% Ib-loaded CANFs coated with poly-AAm.

**Figure 5 polymers-13-01863-f005:**
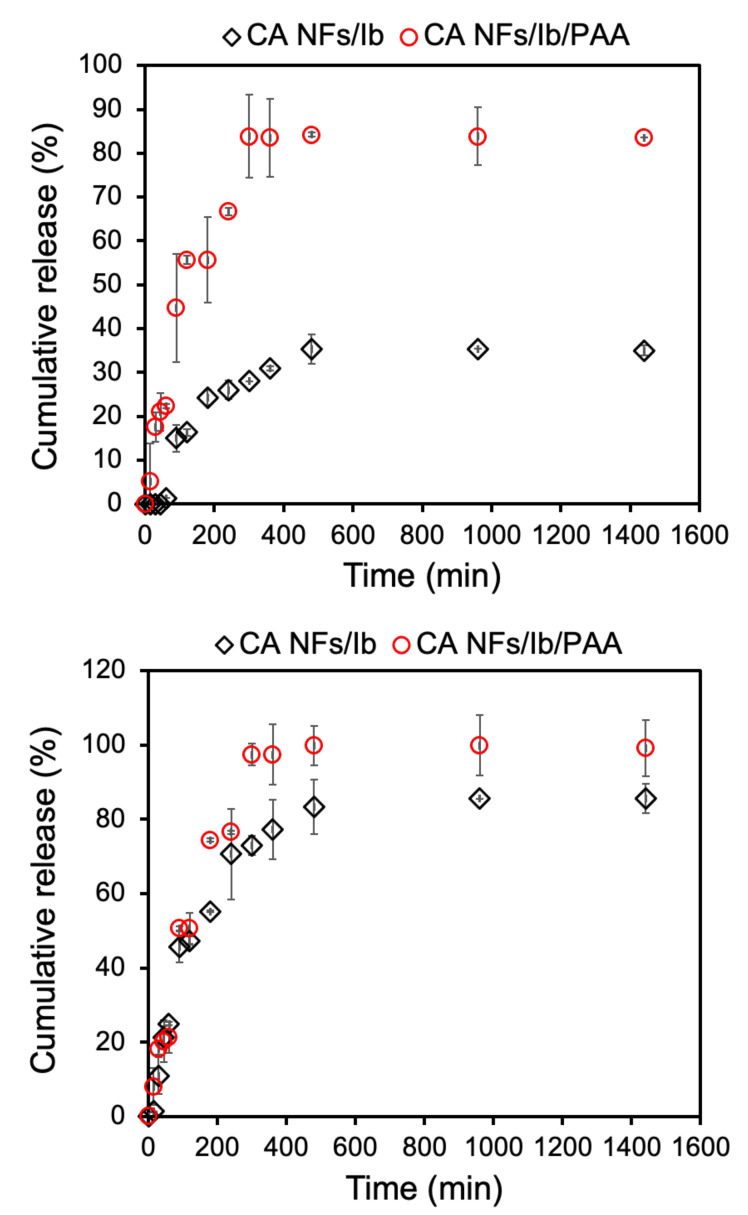
Time independent drug release profile of both non-coated and poly-AAm coated CANFs. (**Top**): 5 wt.% Ib-loaded CANFs; (**Bottom**): 10 wt.% Ib-loaded CANFs (bottom) (Black diamonds: uncoated, red circles: poly-AAm coated). Measurements have been done in triplicate.

**Figure 6 polymers-13-01863-f006:**
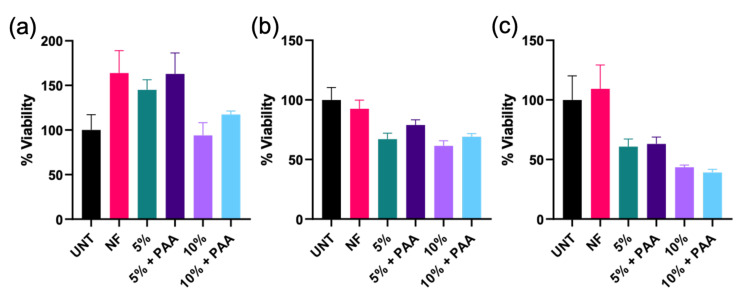
MTS cytotoxicity assays of the electrospun CANFs for 24 h incubation. (**a**) 2 mg/mL, (**b**) 4 mg/mL, and (**c**) 8 mg/mL CANFs. UNT denotes untreated 3T3 adipose cells, NF denotes unmodified CANF, 5% denotes 5 wt.% Ib-loaded CANF, 5% + PAA denotes 5 wt. % Ib-loaded CANF coated with poly-AAm polymer, 10% denotes 10 wt.% Ib-loaded CANF, and 10% + PAA denotes 10 wt. % Ib-loaded CANF coated with poly-AAm polymer. Data reported as mean ± SEM (N = 3).

## Data Availability

The data presented in this study are available on request from the corresponding authors.

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
