# Peer review of "In Situ Photopolymerization of Acrylamide Hydrogel to Coat Cellulose Acetate Nanofibers for Drug Delivery System"

_polymers, 2021, doi:10.3390/polym13111863_

Round 1
Reviewer 1 Report
Thanks for responding.
You still need to mention N and SD.
I have not asked you to repeat the studies.
I just asked to report your statistics.
Author Response
Point 1: Thanks for responding. You still need to mention N and SD. I have not asked you to repeat the studies. I just asked to report your statistics.
Response 1: We repeated the cytotoxicity experiment, and a new Figure was added with N and SD as Figure 6 into the revised manuscript and the old figure was moved to the supporting information.
Reviewer 2 Report
The article seems to be interesting. The research topic undertaken by Authors is worth investigating. The methodology applied and the conclusions drawn by Authors are proper. In general, the main concept of the proposed article is worth noticing. Nonetheless, the work requires some improvements, i.e.:
- Abstract of the paper should be significantly shortened because it is too extensive.
- Section Materials: the short descriptions of the reagents used including e.g. their molecular weight or density as well as their suppliers need to be added.
- Section 2.5.3.: the principle of MTS assay is not clear. Do live cells absorb the MTS reagent whether the reaction with this reagent occurs in the tested environment (as it takes place in the case of e.g. MTT assay)? This should be specified.
- In the first paragraph of Section 3. Authors mentioned about the shortcomings associated with poor drug release behavior which are observed with conventional methods for nanofiber-based drug delivery applications. This issue should be developed. Additionally, the information concerning the conditions of nanofibers drying (line 246) should be added.
- Discussion over the results obtained should be supported by references to other investigations. For example, Authors mentioned during the analysis of XRD results that “For Ibuprofen, the XRD pattern shows high crystalline peaks which matched previously reported patterns in the literature” without indicating any literature references – it is not proper.
- From the editorial point of view, the notation of units in figures should be improved to be consistent. In some cases the unit is given in brackets (e.g. in Figure 2.: wavenumber and temperature) and in some cases not (e.g. in Figure 2.: transmittance and weight loss).
- Figure 6.: there is no information concerning the repeatability of the results obtained (the amount of the measurements performed for each sample).
- Lines 337-340: Authors mentioned that the release profile of bioactive molecules from electrospun fibers can be influenced by biodegradability, hydrophobicity, hydrophilicity, fiber diameter, and configuration. These dependencies should be briefly commented to increase the scientific value of the paper.
Author Response
General comment: The article seems to be interesting. The research topic undertaken by Authors is worth investigating. The methodology applied and the conclusions drawn by Authors are proper. In general, the main concept of the proposed article is worth noticing. Nonetheless, the work requires some improvements, i.e.:
Point 1: Abstract of the paper should be significantly shortened because it is too extensive.
Response 1: The abstract was revised again to be succinct and concise.
Point 2: Section Materials: the short descriptions of the reagents used including e.g. their molecular weight or density as well as their suppliers need to be added.
Response 2: Done.
Point 3: Section 2.5.3.: the principle of MTS assay is not clear. Do live cells absorb the MTS reagent whether the reaction with this reagent occurs in the tested environment (as it takes place in the case of e.g. MTT assay)? This should be specified.
Response 3: It is well known principle for cytotoxicity assessments. Lactate dehydrogenase (LDH) is a cytosolic enzyme present in many different cell types. When the plasma membrane is damaged, LDH is released into the cell culture media. MTS colorimetric reagent then can be used to quantitatively measure LDH released into the media from damaged cells as a biomarker for cellular cytotoxicity. Thus, we measure the dead cells then calculated the final cell viability. The difference between MTT and MTS assays is that MTT has an additional step associated with the solubilization of formazan crystals whereas MTS is not associated with the solubilization of formazan crystals.
Also as in response 1, we repeated this experiment, and a new Figure was added with N and SD as Figure 6 into the revised manuscript and the old figure was moved to the supporting information.
Point 4: In the first paragraph of Section 3. Authors mentioned about the shortcomings associated with poor drug release behavior which are observed with conventional methods for nanofiber-based drug delivery applications. This issue should be developed. Additionally, the information concerning the conditions of nanofibers drying (line 246) should be added.
Response 4: Narrative was added with citations to the first part of the question and the drying method was added too to the revised manuscript.
Point 5: Discussion over the results obtained should be supported by references to other investigations. For example, Authors mentioned during the analysis of XRD results that “For Ibuprofen, the XRD pattern shows high crystalline peaks which matched previously reported patterns in the literature” without indicating any literature references – it is not proper.
Response 5: Citation was added there.
Point 6: From the editorial point of view, the notation of units in figures should be improved to be consistent. In some cases the unit is given in brackets (e.g. in Figure 2.: wavenumber and temperature) and in some cases not (e.g. in Figure 2.: transmittance and weight loss).
Response 6: Done.
Point 7: Figure 6.: there is no information concerning the repeatability of the results obtained (the amount of the measurements performed for each sample).
Response 7: As in response 1, we repeated the cytotoxicity experiment, and a new Figure was added with N and SD as Figure 6 into the revised manuscript and the old figure was moved to the supporting information.
Point 8: Lines 337-340: Authors mentioned that the release profile of bioactive molecules from electrospun fibers can be influenced by biodegradability, hydrophobicity, hydrophilicity, fiber diameter, and configuration. These dependencies should be briefly commented to increase the scientific value of the paper.
Response 8: Narrative with citations was added to improve this part.
This manuscript is a resubmission of an earlier submission. The following is a list of the peer review reports and author responses from that submission.
Round 1
Reviewer 1 Report
Major Issues
In my opinion, drug release studies are meaningless to the light of the authors explanation. In fact, it is quite clear that ibuprofen is extracted from the nanofibers during the polymerization process.
Moreover, there is no evidence that nanofiber are still present into di PAA hydrogel.
Authors stated that they measured the DL after this process but, since they cannot isolate the nanofibers from the crosslinked hydrogel, they cannot state that ibuprofen is still inside the fibers!
In other words, it means that the increase in release rate is related to the fact that ibuprofen is not into the fibers but into the hydrogel. Still, generally in the development of a drug delivery system a prolonged release of the active molecule is desirable. Honestly, in my opinion the ibuprofen release profile from uncoated electrospun fibers is advantageous compared with the coated sample. No burst release can be observed from uncoated sample. Please notice that these results are central for the entire work rationale so in my opinion the manuscript still does not fits the scientific quality of the journal.
Minor comments
There is a lot of superfluous information in the introduction that can be eliminated (e.g. the reference to deacetylation of cellulose acetate or the part relating to cancer imaging)
What is the point of talking about the properties of nanofibrillary scaffolds (high surface / volume ratio) if after the hydrogel formation process the fibrillar structure is completely lost (because it is incorporated)?
What is the advantage of having used an electrospun material for drug incorporation instead of using only the hydrogel?
There are no differences in the IR spectra for drug loaded and drug free samples.
Reviewer 2 Report
Reviewer’s Comments
General comment
- The manuscript entitled "In situ Photopolymerization of Acrylamide Hydrogel to Coat Cellulose Acetate Nanofibers for Drug Delivery System" explored the releasing of gentamycin-incorporated polylactide-polyvinylalcohol function as wound dressing. It was a good topic in wound dressing application but I would like to recommend acceptance with major revision prior to journal acceptance. Below are the related concerns should be addressed:
Title
- The title seems to be genuine and clear for the aim of this study.
Abstract
- Understandable and concise abstract explaining the aim and output of this particular study. However, the abstract structure needs to be improved for the result section with provide more “values” rather than subjective statement only.
Introduction
- Why this manuscript had red and black font color text? Edited version or rejected by other journal? Please clarify the content purity. Thanks.
Experimental section
- Looks good and appropriate experimental design.
Results & Discussion
- Acceptable at the current form. But;
- Figure 5 - need to be improve, better in graph line
- Figure 6 - no error bars at all? no significant difference? Should be label 4 & 8 mg/ml
Conclusion
Please improve the conclusion part to address the final output.
Research Ethic Approval
No external research ethic for this study? No funding resource??

Reviewer 3 Report
Thank you, it was an interesting paper.
I have few comments that need to be addressed before publication.
1- when you mention "efficient targeted therapeutic carriers for delivering nucleic acids [36,37], proteins [38], microorganisms [39], and stem cells [40],”
Reference 40 is not even related to stem cell delivery. You should cite at least the works that are related to electrospinning stem cells like below:
Nosoudi, Nasim, et al. "Electrospinning live cells using Gelatin and Pullulan." Bioengineering 7.1 (2020): 21.
2-Can you please show your electrospinning setting? What's a metal capillary? Is it a needle?
3-What is your rationale for using 3T3 Adipose cells? Adipose TE?
4-You have some big chunk of polymer in the blank electrospun fibers as seen in b. Were the drug-loaded ones the same? You don’t show the same magnification for drug-loaded ones so it's hard to tell!
5-Why don’t we see a complete drug release from the uncoated samples? please explain
6-When you say” For CANFs dosed at a concentration of 4mg/mL” does it mean you used 4 mg of fibers incubated with 1 ml of media? If so please change the wording.
7-There is no standard deviation in your viability chart, does it mean that you had n=1? If not then where is the statistical analysis? Without that, you can't really conclude anything for viability
8-You have mentioned that “Overall, the developed CANFs exhibited excellent biocompatibility in comparison with other polymeric materials at the same concentrations” which materials are you referring to?